# Basic Pathogenic Mechanisms and Epigenetic Players Promoted by Extracellular Vesicles in Vascular Damage

**DOI:** 10.3390/ijms24087509

**Published:** 2023-04-19

**Authors:** Concetta Schiano, Carolina Balbi, Filomena de Nigris, Claudio Napoli

**Affiliations:** 1Department of Advanced Medical and Surgical Sciences (DAMSS), University of Campania Luigi Vanvitelli, 80138 Naples, Italy; concetta.schiano@unicampania.it; 2Laboratory of Cellular and Molecular Cardiology, Cardiocentro Ticino Institute, 6807 Taverne-Torricella, Switzerland; 3Department of Precision Medicine, University of Campania Luigi Vanvitelli, 80138 Naples, Italy; 4Clinical Department of Internal Medicine and Specialistic Units, Division of Clinical Immunology and Immunohematology, Transfusion Medicine and Transplant Immunology (SIMT), Azienda Universitaria Policlinico (AOU), 80138 Naples, Italy

**Keywords:** extracellular vesicle, exosome, vascular damage, atherosclerosis, biomarkers, cancer

## Abstract

Both progression from the early pathogenic events to clinically manifest cardiovascular diseases (CVD) and cancer impact the integrity of the vascular system. Pathological vascular modifications are affected by interplay between endothelial cells and their microenvironment. Soluble factors, extracellular matrix molecules and extracellular vesicles (EVs) are emerging determinants of this network that trigger specific signals in target cells. EVs have gained attention as package of molecules with epigenetic reversible activity causing functional vascular changes, but their mechanisms are not well understood. Valuable insights have been provided by recent clinical studies, including the investigation of EVs as potential biomarkers of these diseases. In this paper, we review the role and the mechanism of exosomal epigenetic molecules during the vascular remodeling in coronary heart disease as well as in cancer-associated neoangiogenesis.

## 1. Introduction

The vasculature controls homeostasis and the functional integration of tissues and organ systems [1,2]. Hemodynamic and bioactive stimuli stimulate the vasculature, which responds with multiple structural and phenotypic changes that are well described during the onset and progression both of cardiovascular diseases (CVDs) and cancer [1,2,3]. Starting from fatty streaks, cell components of the vascular wall change their function, contributing to the progression of atheromatous plaques, chronic inflammation and clinical manifestations of coronary heart disease (CHD) [4,5,6,7,8,9]. Endothelial adaptative remodeling also occurs during cancer growth, representing a key event during the transition from local to systemic and metastasis dissemination [10]. Different metabolic and epigenetic signaling between endothelial cells and the microenvironment are crucial to the initiation and progression of pathological vascular modifications [1,2,3,11]. Therefore, investigation of the pathophysiology of the vascular remodeling mechanism both in CVDs and cancer is of great value for finding new disease targets and biomarkers. A central role in vasculature modifications is the crosstalk between several cell types necessary to transfer information from one site to another. An emerging body of evidence implies that extracellular vesicles (EVs), which circulate in biological fluids, are vehicles of a variety of molecules to recipient endothelial cells during CVD and cancer growth [12,13,14,15,16,17,18]. However, the role of EVs and their epigenetic influences in the pathogenesis of vascular damage are poorly explored. Here, we discuss the epigenetic effects exerted by EVs on the vasculature both in coronary heart disease (CHD) and cancer.

## 2. EVs Biology and Function

According to the International Society for Extracellular Vesicles (ISEV) (https://www.isev.org, accessed on 15 April 2023), vesicles include both small and medium/large particles, also known as exosomes (Exo) and microvesicles (MVs), which differ by size and effect on biogenesis [19,20]. The literature reports that EVs can be isolated from biological fluids, such as serum, plasma, urine and amniotic fluid, as well as from the conditioned medium of cells cultured and treated in vitro [12,13,16]. EVs can provide “circulating information” to promote intercellular communication [20]. However, since their number and molecular composition can differ considerably, depending on the micro-environmental and pathophysiological status of their cellular origin, their purity and the method of preparation are very important [21,22]. The EVs’ cargo is known to include proteins and nucleic acids such as DNA, mRNA and a variety of non-coding RNAs (ncRNAs) that, when released in target cells, differentially regulate gene expression [23,24,25]. Exosomes not only regulate physiological states, such as tissue regeneration, immune surveillance and stem cell plasticity [26,27,28,29], but also participate in the pathophysiology of CVD, neurodegenerative diseases and malignant tumors [30,31,32] and have potential value for precise diagnosis, prognosis assessment and disease treatment. Moreover, given their biological role, EVs can affect the pathogenesis of CVD and tumor progression [33,34,35]. Specific EVs containing microRNAs (miRNAs), long non-coding RNAs (lncRNAs) and circRNAs, may recruit DNA methylase (DNAm) or histone deacetylase enzymes to induce epigenetic changes in target cells [5]. Growing evidence indicates that epigenetic mechanisms are closely related to cardiovascular disease development and cancer neoangiogenesis [36,37,38]. Herein, we review recent findings on the interplay between the molecular cargo of EVs that induces epigenetic modifications in vessel during both CVD and cancer vascularization. 

## 3. The Epigenetic Role of EVs in Cardiovascular Damage

### 3.1. EVs and Peripheral Endothelial Dysfunction 

Epigenetic changes contribute to the development of CVDs and begin with progressive alterations of the endothelial tissue [3,7]. Endothelial dysfunction is characterized by impaired vascular function, loss of antithrombotic function, excessive proliferation of vascular smooth muscle cells (VSMCs), platelet activation and macrophage migration [9,39]. In endothelial dysfunctional cells, intercellular and vascular cell adhesion molecule-1 (ICAM-1 and VCAM-1) are overexpressed and promote the leukocytes’ adhesion [40]. Several aberrant epigenetic modifiers, mainly miRNAs and lncRNAs, are transported by EVs on dysfunctional endothelial cells (ECs) that modify gene expression and contribute to the acceleration of vascular disease to advanced atherosclerosis [41,42] (Figure 1). 

Ding Yu Lee and colleagues demonstrated that the downregulation of miR-10a in both the circulation (serum) and aortic endothelium is highly associated with atherogenesis. miR-10a reduced leukocyte infiltration through GATA6/VCAM-1 signaling pathway, thereby decreasing plaque formation. Thus, this study showed that miR-10a downregulation is associated with atherosclerosis and its induction could prove to be a novel therapeutic target for this disease [43]. Evidence also demonstrated that several miRNAs regulate the leukocytes’ infiltration into the endothelium vessel. One example is miR-155-EVs from VSMCs that target ECs down-regulate integrin proteins and tight junctions. These EVs generate leaky membranes, favor leukocytes and other immune cells migration into sub-endothelium where macrophages can maturate in the foam cells [3,7,40,44]. Moreover, data showed that miRNA-155- and miRNA-223-EVs may also target mononuclear cells, favoring LDL accumulation and their transformation in damaged endothelium [45,46]. Hall and colleagues outlined the importance of an EV-derived circRNA, named Circ-Lpr6, for vascular pathogenesis in both animal and human models. This novel circRNA acts as a sponge for miR-145, thereby regulating its downstream targets. Molecular studies demonstrated that Circ-Lpr6 impairs miR-145-mediated regulation of VSMCs differentiation, migration and proliferation. Furthermore, its overexpression resulted in a decreased intimal area in mouse aortas. Therefore, Circ-Lrp6 could be required to counterbalance miR-145 into VSMCs, providing further evidence that ncRNAs are an important target in this disease, as the ratio between Circ-Lpr6 bound to miR-145 versus unbound plays a role in vascular pathogenesis [47]. Another feature of endothelial dysfunctional vessels is the alteration of the coagulation pathway due to abnormal platelets activation. EVs released following this stimulus are found in chemokine RANTES (normal activation-regulated, expressed, and presumably secreted T cell), thus promoting the neutrophils’ adhesion and migration, starting inflammation [48].

### 3.2. EVs in Inflammation and Calcification

Cellular damage plays a critical role in the inflammatory response, delivering a high amount of EVs containing pro- and anti-inflammatory signals to target cells [3,7,43]. Although ncRNAs are not classically considered as direct epigenetic modificators, they might have a role in such events. Protective mechanisms are activated by miRNA-126, miRNA-210 and miRNA-214 promoting angiogenesis and vasculogenesis [48,49,50]. In particular, miRNA126 increased the expression of chemokine 12 ligand (CCL12), reduced macrophage entry into the vascular wall and protected the vascular endothelium [51]. A supportive role for this action mechanism derived from the observation that the silencing of miRNA-126 caused leaky vessels, hemorrhage of blood from arterial vessels and partial embryonic lethality. Silencing also induced a loss of vascular integrity and defects in ECs proliferation and angiogenesis [52]. Although miRNA-214 targeting damaged ECs, it also promoted angiogenesis and thus favored the restoration of vessel integrity [50]. The contribution of EVs to endothelial inflammation has been reported to be due to DC-derived EVs that activated the TNF-α/NF-κβ pathway, contributing to the progression of atherosclerosis in animal model [53]. In response to inflammatory stimuli, ECs transferred EV-miR-92a-3p to VSMC, resulting in cell proliferation, while EV miR-146a inhibited their migration, further exacerbating the inflammatory response [54,55,56]. The RNCR3 ncRNA released from ECs in aortic atherosclerotic lesions showed a protective role, regulating miR-185-5p in VSMC target cells. Specifically, miR-185-5p favored the interaction between RNCR3 and Klf2, promoting ECs vasoprotection [57]. In vivo knockdown of RNCR3 worsens atherosclerosis, hypercholesterolemia and inflammation [57]. Thus, EV-transported ncRNAs have a key role in the onset of CVDs. EVs are also involved in the progression of vascular calcification [58]. Analysis of murine and human vessels showed that macrophage-derived EVs promoted intimal microcalcification [59]. Furthermore, the enrichment of these EVs with the calcium-binding protein S100A9 facilitated the calcification progression together with VSMC-derived EVs [59]. In response to this stimulus, miRNA150-EVs promoted vascular maintenance by activating the endothelial vascular endothelial growth factor (VEGF)-A/VEGFR/PI3K/Akt pathway [60]. Although little data are available on EVs regulating apoptosis during endothelial damage, it was demonstrated that HUVECs treated with macrophage-derived EVs activate NF-κB and TLR4 signaling pathways, causing ECs apoptosis [53].

### 3.3. EVs and CHD

Inflammation and calcification have interconnected roles in the pathophysiology of CHD and acute coronary syndrome (ACS) [3,7,11]. For the first time, the transcriptomic analysis of EVs from ACS patients revealed the presence of mRNA coding for epigenetic enzymes, such as DNA methyltransferases (DNMTs) [61]. Notably, ACS-derived EVs absorbed on PBMCs from healthy subjects modulated the expression of several genes, presumably via epigenetic mechanism in agreement with previously reported [62,63,64,65,66]. In particular, EVs induced methylome changes promoting a significant activation of Synaptosin (SYP), Cell Adhesion Molecule L1 Like (CHL1) and SH2 Domain Containing Adaptor Protein B (SHB) genes [61]. These findings underlined the epigenetic contribution in the role of EVs in ACS.

In the clinical practice, CHD diagnosis is made by coronary CT scan [7,67] and coronary angiography [7,68]. However, it is still unclear whether EVs could be a tool to improve CHD risk estimation future investigations should be conducted to combine EVs and clinical parameters. Shi and colleagues demonstrated that human blood samples from healthy subjects have 86% more EVs compared to CHD patients [69,70]. Therefore, both EV count and content are of potential interest as disease-specific biomarkers [71,72,73,74,75].

Nowadays, it is known that, in the cardiovascular system, EVs also play important epigenetic-indirect roles in the acute post-myocardial infarction (MI) damage and repair mechanisms inducing myocardial remodeling and cardiac regeneration [76]. Recently, echocardiographic experiments had shown that exosomes derived from the mesenchymal stem cell of adipose tissue (ADSC) could improve left ventricular ejection fraction, whereas their administration could significantly alleviate MI-induced cardiac fibrosis [77]. Additionally, ADSC-exosomes treatment has been shown to reduce cardiomyocyte apoptosis, increasing angiogenesis. Molecular experiments demonstrated that ADSCs-derived exosomes can promote microvascular ECs proliferation and migration and inhibit cardiomyocytes apoptosis through miRNA-205. This assumption was confirmed via the transfection of ADSC-derived exosomes into MI-induced mice and observing a decrease in cardiac fibrosis, an increase in angiogenesis, and an improvement of cardiac function [77]. Cardiac fibroblast-derived EVs modulate cardiac remodeling, favoring intercellular communication between cardiomyocytes (CM) and cardiac fibroblasts [78,79]. Consistently, exosomal miRNA-21 CM-derived, reduced the expression of sarcoplasmic protein sorbin 2 containing SH3 domain (SORBS2) and PDZ and LIM domain 5 (PDLIM5), producing an increase in cell size and stimulating cardiomyocyte hypertrophy [80,81]. Moreover, miRNA-21 promoted the cardiac fibrosis inducing the endothelial-to-mesenchymal cell transition [82] and altered expression of superoxide dismutase (SOD) in HF patients [83,84]. Increasing research has attracted attention to the roles of EVs-ncRNAs as potential diagnostic biomarkers. Other than miRNAs, several studies have indicated that lncRNAs significantly regulate fibrosis, thereby having a direct effect on ECM gene expression, the TGF-β signaling pathway and the proliferation of fibroblasts or transition to myofibroblast [23,85,86,87]. These effects are also proposed to be mediated by paracrine communication of EVs between donor and recipient cells, especially in cardiomyocytes and fibroblasts [23]. Exosomes-containing lncRNA ZFAS1 could induce cardiac fibrosis via the Wnt4/β-catenin signal pathway by sponging miR-4711-5p in cardiac fibroblasts [85]. LncRNA MIAT is up-regulated in serum-derived EVs from atrial fibrillation (AF) patients. MIAT aggravated the atrial remodeling and promoted AF by binding with miR-485-5p [86]. Neat1 is obviously up-regulated by P53 and HIF2A in cardiomyocytes in response to hypoxia and is enriched in cardiomyocyte-derived exosomes. Neat1 is essential for cell survival and fibroblast functions. This finding was confirmed via genetic knockout of Neat1 that impaired cardiac function during MI [87]. Although currently studies aim at the analysis of EVs from human samples in AF, HF and MI repair, large studies need to be performed to determine their utility in the clinical routine [88]. In addition to all the findings on basic mechanisms involving EVs, various clinical studies have been initiated to date. Some of these are still at the recruitment stage, and the few completed studies have not yet published their first results (NCT03660683, NCT04142138, NCT03034265, NCT03984006, NCT02822131, NCT00331331, NCT03837470 and NCT04266639). All studies listed in Table 1 will investigate the general cargo of EVs from different biological fluids and tissues, but none have provided help to investigating specific molecules. Trials are evaluating the role of EVs as carriers of anti-inflammatory signal or drug delivery systems. However, these are still few, and their results are not fully available (Table 1).

## 4. Basic Mechanisms and Epigenetic Role of EVs in Tumor Vascularization

### 4.1. EVs in Neoangiogenesis

Initiation of angiogenesis is an important early event in premalignant lesions to supply nutrients and oxygen [1,9] (Figure 2). 

EVs such as extracellular organelles elicit signaling between tumor and vascular cells carrying functional and epigenetic modifications to adapt the microenvironment to tumor growth conditions (Figure 2). Tumor hypoxia is the most potent stimulus of EVs biogenesis [89]. EV content may activate the proliferative and migratory status of ECs, favoring neo-vessel formation via different mechanisms, including epigenetic ones [90] (Figure 2). The canonical VEGF (or Notch signaling pathway) was described as being activated by EVs from different tumor types because ECs promoted sprouting and proliferation [91,92,93,94] (Figure 2). Other metabolic programs have also been proven to influence the migration of ECs [16]. We reported that EVs containing EDIL3 activate a mitochondrial and vesicular trafficking program through the purinergic receptor 4 in normal quiescent ECs. This stimulus promotes endothelial cell survival, proliferation, motility and organization into capillary-like structures in vitro and in vivo [16]. 

EVs may also regulate endothelial function via epigenetic mechanisms. In preclinical and clinical settings, a variety of ncRNAs and long noncoding RNA (lncRNA) transcripts packaged in EVs have been shown to be capable of modifying gene expression without altering the DNA sequence [95]. Generally, EVs-miRNAs are transferred into ECs, where they are post-transcriptionally downregulating antiangiogenic genes, promoting growth and vessel sprouting. However, the role of individual exosomal miRNAs is dependent on tumor histology and stage. In squamous cell carcinoma (ESCC), miR-181b-5p predicts a poorer overall survival. It was also shown to be involved in vessel formation by promoting endothelial growth via AKT targeting endothelial transcript for PTEN and PHLPP2 [96]. In multiple myeloma, the number of EVs carrying miR-135b is significantly increased compared to normal individuals. Transfer of this miRNA in ECs downregulates the inhibitor of hypoxia-inducible factor 1 (HIF-1), resulting in the formation of endothelial tubes through the HIF signaling pathway [97]. MiR-9-EVs from glioma play a pivotal role in both pathogenesis and vascularization. When miR-9 is absorbed by ECs, the increased neoangiogenesis activates multiple pathways, among them COL18A1, Thrombospondin-2 (THBS2), Protein patched homolog 1 (PTCH1) and prolyl hydroxylase 3 (PHD3) via MYC and OCT4 [98]. One of the most common miRNAs associated with poor prognosis of several cancer types is miR-210 [99,100]. Exosomal miR-210 uptake from ECs promotes angiogenesis by targeting specific processes depending on the tumor microenvironment [100]. The chromatin epigenetic regulation of proangiogenic genes has also been attributed to different lncRNA packaged in EVs, although the mechanism is not fully elucidated [101,102,103] (Figure 2). LncRNAs can work as molecular scaffold, assembling several chromatin regulator proteins and reshaping the DNA structure in the nucleus and thus favoring the transcription or silencing of target genes [104]. In other cases, lncRNAs can act as ceRNAs or sponges, sequestering endogenous miRNAs and inhibiting their action on target. For example, gastric cancer exosomal lncRNA PVT1 was reported to have the capacity to activate proliferation of ECs by forming a complex with polycomb repressive complex 2 (PRC2), the major histone methyltransferase on the STAT3 promoter, activating the transcription of the VEGFAs [105]. Several exosomal lncRNAs and proteins have been identified that originate from non-small cells lung cancer (NSCLC), some of which have a role in neoangiogenesis, e.g., lncRNA-p21 and Ubiquitin-fold modifier conjugating enzyme 1 (UFC1) protein [105,106,107]. The mechanism identified for UFC1 revealed the binding to the enhancer of Zeste 2 (EZH2) protein in the polycomb repressor complex 2 (PRC2) and their accumulation at the promoter region of the PTEN gene. This complex catalyzes the trimethylation of H3K27, the inhibition of PTEN expression and the activation of angiogenesis via AKT. UFC1 knockdown inhibited NSCLC growth in a mouse xenograft tumor model [107]. Another pro-angiogenic mechanism is downstream Angiopoietin 2/Tyrosin kinase endothelial receptor 2 (Ang2/Tie2). This pathway is activated by lncRNA EPIC1 in NSCLC and promotes channel formation and the proliferation of ECs [108]. Furthermore, the switch of ECs towards the proliferative phenotype was attributed to exosomal MANTIS. This lncRNA influenced the epigenetic regulation of angiogenic sprouting and alignment of ECs, modifying the chromatin complex, SWI/SNF (Switch/Sucrose non-fermentable), to facilitate the access to RNA polymerase II machinery and gene transcription [109]. 

Different mechanisms have been attributed to metastasis-associated lung adenocarcinoma transcript 1 (MALAT-1), which may promote angiogenesis in endothelial ovarian carcinoma and in lung cancer [110,111]. Studies have demonstrated the influence of MALAT-1 and other lncRNAs in the complex interplay among lncRNA-miRNA-mRNA governing the sequestration and inactivation of miRNAs. For example, lncNKX2–1 from gastric cancer was reported to increase tumor growth and angiogenesis in both in vitro and in vivo models to direct sponge miR-145–5p, which upregulates Serpin Family E member 1 (SERPINE1) and vascular endothelial growth factor receptor (VEGFR) 2 signaling pathways [112]. A similar mechanism was also defined for receptor activity-modifying protein 2 (RAMP2-AS1) from chondrosarcoma, Urothelial cancer associated (UCA1) for pancreatic cancer, Small nucleolar RNA host gene 1 (SNHG1) from breast [112,113,114], RNA CCAT2 in exosomes from gliomas [115,116] and LINC00707 from cervical or bladder cancer [117,118]. Taken together, these data indicated that EVs have a role in tumor neoangiogenesis through epigenetic mechanisms. Emerging research is evaluating exosomes and their content as potential cancer biomarkers in clinical setting (Table 2). A survey on ClinicalTrials.gov (https://clinicaltrials.gov/, accessed on 15 April 2023) shows a total of 88 trials, of which 60 (68%) EVs’ cargo from different source are used as biomarkers of disease or response to therapy applications. In the case of exosome therapy, as a drug-delivery system, seven trials (8%) have been registered and one clinical trial (1.72%) is related to exosome as vaccine study. Although no definitive data are available, a manufacturing practice (GMP) as exosome production method and purification methods is present in the current trial. Furthermore, the European Medicines Agency (https://www.ema.europa.eu/en, accessed on 15 April 2023) has released scientific recommendations on classification of advanced therapy medicinal products. This nascent field of biomedicine offers new opportunities for the treatment of diseases and dysfunctions of the human body. In addition, it provides guidelines, recommendations, for researchers. However, the transition of EVs from basic research to clinical application is still premature.

### 4.2. EVs and Vascular Alterations in Tumor Progression

The disruption of vascular integrity and the consequent increase in vascular permeability allows tumor cells to penetrate the vasculature and metastasize. EVs secreted by cancer cells during tumor growth play an important role in vascular leakage (Figure 2). Studies have demonstrated that EVs from metastatic cancer cells may modify ECs’ permeability by altering cytoskeletal-associated proteins or downregulating the RhoA/ROCK pathway [119]. EVs containing miRNAs (including miR-200c, miR-141 and miR-429), which correlated with a poor prognosis of colon cancer, were demonstrated to downregulate ZEB protein and modify the membrane permeability of ECs [120]. Similarly, EVs carrying miR-25-3p, more frequent in patients with metastasis than in those without, were demonstrated to induce vascular leakage and enhance colon cancer metastasis [121]. The same is true for EVs miR-105 in breast cancer [122]. Furthermore, EVs miR-939 correlated with poor prognosis of gastric cancer and increased the permeability of ECs by downregulating the transcription of VE-cadherin [123]. A similar mechanism was reported for LncX26 in gastric cancer [124]. Additionally, the loosening of endothelial integrity and ultimately inducing colon cancer metastasis were attributed to post-transcriptional down regulation of p120-catenin (p120) mediated by transferring of exosomal miR-27b-3p from colon cancer [125] or miR-638 from HCC [126]. In NSCL cancer patients, exosomal miR-375-3p positively correlated with patient TNM stages. In vivo, miR-375-3p-enriched exosomes destroyed the endothelial structure of lung, liver and brain tissues of mice, leading to increased blood vessel permeability. Exosomal miR-375-3p internalized by breaking the tight junction of ECs through negative regulation of claudin-1 expression [127]. Another important pathway of EVs systemic activity is the degradation of extracellular matrix (ECM), favoring tumor growth and preparation of the metastasis niche. In particular, MMP-1 in EVs derived from cancer cells is a representative indicator of ECM degradation [128] (Figure 2). Evidence shows that the pharmacological reduction of EVs decreases the secretion of MMPs, resulting in an inhibition of tumor growth and metastasis [128,129]. This pathway is epigenetically regulated by exosomal lnc-MMP2-2 released by NSLC. The lncRNA internalized into ECs competed with miR-1207-5p to downregulate EPB41L5 and increased vascular permeability, promoting brain metastasis [130]. These data indicate a key role of EVs in damaging the endothelium and promoting intravasation of tumor cells and extravasation at distal sites (Figure 2). Moreover, considering the epigenetic roles of some ncRNAs and their capacity to control gene expression, the development of epigenetic-related drugs with higher specificity and fewer side effects would be a possible future approach.

### 4.3. EVs in Metastatic Niche: And Other Types of Vessels 

Tumor vasculatures increase the tumor capability to metastasize by acquiring mesenchymal characteristics [131,132]. Both normal and tumor endothelial ECs are capable of acquiring a mesenchymal phenotype, increasing the tumor’s ability to metastasize. TECs comparing to ECs exhibiting cytogenetic abnormalities, such as aneuploid chromosomes, are differ metabolically, more resistant to starvation and release more angiocrine molecules and EVs [133,134]. The presence of TEC is associated with conversion of indolent tumor to more aggressive phenotype, and their presence was correlated with reactivation of dormant CD44v6+ tumor stem cells [135,136]. ECs and TECs within the vasculature gain a mesenchymal phenotype (vimentin alfa-smooth muscle actin positive), in some tumor types, shed from the endothelial lining layer of neoplastic vasculature into peripheral blood to the colonizing distal organ [135,136,137,138]. The mechanism is not fully understood. Exosomal Annexin A1 is one of the triggers inducing endothelial-to-mesenchymal transition (EndEMT) and activating RAC1/PAK2 in the gastric cancer model, whereas TGF-β on the surface of EVs triggers the same phenotypic switch in melanoma [139,140,141]. Moreover, data from EVs containing miR-92a-3p, derived from colon cancer cells, showed upregulation of mesenchymal markers, such as snail and vimentin, and downregulated the tight junction marker ZO-1 in HUVECs referred as “partial-EndoMT” [142]. EVs released from tumor cells during hypoxia participate in the EndMT switch of TECs, particularly EVs loaded with lysyl oxidase-like 2 protein (LOXL2) [143]. LOXL2 is a member of the LOX family and regulates extracellular matrix (ECM) remodeling, angiogenesis and premetastatic niche formation. LOXL2 acts as deaminase enzyme into the nucleus contributing to transcription, alternative splicing and miRNA regulation. All these pathways contribute to EndoEMT transition, proliferation and migration of ECs and TECs to distant organs and vasculogenic mimicry mechanism (Figure 2) [143]. Exosomal lncRNA antisense transcript of GATA6 (GATA6-AS) has also been reported to bind LOXL2, which regulates endothelial gene expression underlying EndoEMT transition [144]. The authors demonstrated that GATA6-AS associated with LOXL2 activates H3K4me3 marks and positively regulates the angiogenesis-related genes periostin and cyclooxygenase-1 in ECs. [144]. According to recent studies a multitude of other forms of neovascularization have been observed in cancer settings, including vascular looping and splitting (intussusception), vasculogenic mimicry and remodeling of larger ‘feeding’ vessels (arterio/venogenesis). An epigenetic mechanism mediated by EVs was also involved in vascular mimicry (Figure 2). Here, tumor cells undergoing epithelial-mesenchymal transition (EMT) form microvascular channels contributing to tumor vascularization [145]. These vessel types are negative to CD31 antigen and insensitive to anti angiogenic therapy. The mechanism involved the transferring of EVs containing lncRNAs into target cells, where they act as sponge of selective miRNAs or as scaffold for chromatin remodeling [145,146,147,148,149,150,151]. These mechanisms have been described for HULC in glioblastoma MALAT-1 in gastric cancer and in lung cancer Linc0155 [152,153,154]. Vasculogenic cooption (VCO) is another way by which tumor cells incorporate vessels from normal tissue to gain access to blood supply and growth locally (Figure 2). It is reported that tumor cells can migrate along blood vessels beyond the advancing front of the tumor and may spread to distant metastases [143]. Vessel cooption was initially described in gliomas and lung metastasis [155,156]. Then, VCO was reported in breast cancer metastasis and gastric metastasis from colon [143]. The mechanism of VCO is mediated by VEGF, but it is reasonable to think that EVs rich in VEGF are also involved [155].

## 5. Conclusions and Perspectives

Some pathogenic aspects of cancer and CHD are defined by the pathophysiology of vascular damage induced by key events in which EVs might play a role. EV counts, types and contents tend to vary, depending on collection methods. By analyzing literature data, we found that only three ncRNAs, specifically, miR-92a-3p, miR-145 and miR-210, were involved in vascular damage, both in the context of CVDs and in tumor neoangiogenesis, although with different mechanisms, underling the possible common pathway. Epigenetic tissue-specific reversible changes induced by exosomal cargo could play a role in the initiation and progression of disease complications. In terms of biomarkers, exosomal ncRNAs could be superior compared to non-exosomal ncRNAs for their specify in methods preparation, the protection from degradation and stability. Moreover, exosomal ncRNAs exist in considerable amounts in different body fluids that can be non-invasively sampled. Considering the epigenetic reversible roles of some ncRNAs, the establishment of novel approaches for modulating their expression or counteract their action could be a future goal [18]. To this end, in the cardiovascular field, 28 clinical studies were started which include exosome analyses (Table 1). In the field of oncology, 88 trials have begun in order to investigate the potential prognostic role of EVs. The most promising application appears to be biomarkers for gastrointestinal cancers (Table 2). However, larger, randomized, prospectively enrolling multicenter trials are necessary to identify EVs as potential clinical biomarkers to better guide clinical treatments.

## Figures and Tables

**Figure 1 ijms-24-07509-f001:**
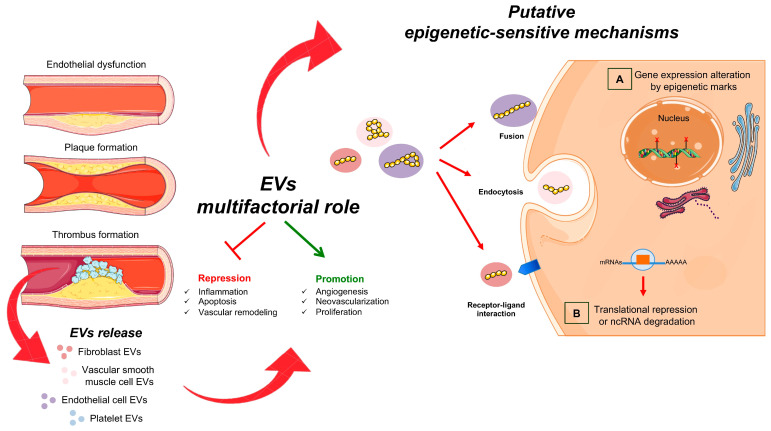
Extracellular vesicles and vascular damage. Schematic representation of atherosclerotic EVs. They are produced by the different cell types involved in the mechanism of plaque formation. Harmful EVs can promote atherogenesis by altering vascular integrity, increasing inflammatory response and promoting thrombus formation. On the other hand, beneficial EVs, containing specific ncRNAs, can play athero-protective roles by promoting the repair of damaged endothelium and repressing the activation of inflammatory cells. The EVs interact with target cells via direct fusion, endocytosis or the binding of surface proteins. After this invagination, cytosolic proteins or RNA content-EVs transfer to the extracellular space. In the target cells, ncRNAs (such as lncRNAs, miRNAs, circRNAs) can (A) control nuclear gene expression via epigenetic changes and/or (B) affect protein function.

**Figure 2 ijms-24-07509-f002:**
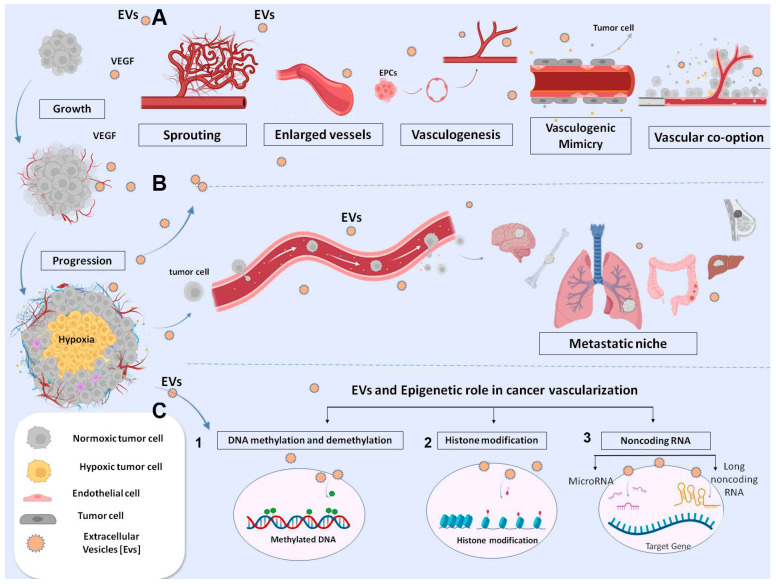
Schematic representation of EVs involved in different mechanisms of tumor angiogenesis. Tumor cell-derived exosomes carrying biological molecules are involved in various angiogenic mechanisms: (**A**) sprouting angiogenesis, vessel enlargement, arteriogenesis, vascular mimicry and vessel-cooption favoring tumor growth and pre-metastasis niche formation; (**B**) Exosomes from malignant cells released locally or at distant sites can induce vessel modifications, with endothelial damage favoring intravasation. Triggering endothelial-mesenchymal transition, which favors endothelial migration at the distal site. Stimulation of extravasation at distant pre-metastatic niche; and (**C**) Schematic representation of epigenetic mechanisms: 1 DNA methylation; 2 Histone modifications; 3 non-coding RNA targets intracellular mRNA.

**Table 1 ijms-24-07509-t001:** Clinical studies of EVs’ evaluation as diagnostic biomarker for CVD treatment.

Number	NCT Number	Status	Conditions	Study Type	Population (Participants Number)	Outcome
1	NCT03660683	Terminated	Diabetes Mellitus, Type 2 Cardiovascular Diseases	Interventional (Dapagliflozin 10 mg)	15	Evaluation of exosomes released from kidney podocyte after treatment, as indicator of kidney podocyte health, via urine assay
2	NCT04142138	Completed	Prehypertension	Interventional (DASH)	9	Characterization of urine exosome protein abundance pattern during nutritional changes, shifting from a “westernized diet” to a DASH diet
3	NCT03478410	Recruiting	AF	Interventional (Epicardial fat biopsy)	35	Quantification of epicardial fat-derived exosomes in patients with and without AF
4	NCT04356300	Not yet recruiting	MODS after surgical repaired for ATAAD	Interventional (MSC-exosome administration intravenously to patients once a day for two weeks)	60	Intravenously treatment with MSC-derived exosomes immediately after ascending aortic replacement combined with open placement of triple branched stent graft
5	NCT05669144	Recruiting	MI Myocardial Ischemia Myocardial Stunning	Interventional (Intracoronary and intramyocardial injection of exosomes)	20	Mitochondria and MSC-derived exosomes evaluation
6	NCT03034265	Completed	Hypertension	Observational	24	Quantification of urinary exosomal sodium channels
7	NCT03384433	Unknown status	Cerebrovascular Disorders	Interventional (Intravenous administration of allogenic MSC-generated exosomes transfected by miR-124 post-stroke)	5	Intravenous administration of allogenic MSC-generated exosomes transfected by miR-124 post-stroke Evaluation of disability improvement of patients with acute ischemic stroke after exosome administration
8	NCT03984006	Completed	Autoimmune thyroid disease Heart Failure	Observational	5	Urinary exosomal NT-proBNP detection to find earlier predicting biomarkers for heart dysfunction
9	NCT02822131	Completed	Hypertension	Interventional (High phosphate diet)	10	Urine exosome evaluation after high phosphate intake
10	NCT05490173	Not yet recruiting	Premature Birth Extreme Prematurity Preterm Intraventricular Hemorrhage Hypoxia-Ischemia, Cerebral Neurodevelopmental Disorders	Interventional (MSC-derived exosomes intranasal administration)	10	EVs effect investigation through analysis of perinatal brain injury biomarkers (S-100, NSE, EPO) and mRNA
11	NCT05035134	Recruiting	Intracerebral Hemorrhage Circulating Exosomes	Observational	300	Circulating exosomes evaluation by RNA and proteome sequencing
12	NCT04127591	Unknown status	MI	Observational	10	Peripheral plasma exosome miRNA evaluation
13	NCT05326724	Recruiting	Post-stroke Dementia Acupuncture	Interventional (Acupuncture)	30	Quantification of acupuncture-induced exosomes
14	NCT00331331	Completed	Uveitis Vasculitis Ocular Inflammatory Disease	Observational	147	Vitreus exosome evaluation
15	NCT05243368	Not yet recruiting	CCU	Interventional (MSC-derived exosome treatment)	30	Evaluation of microenvironment favoring tissue regeneration after MSC-derived exosome treatment
16	NCT03837470	Completed	HFpEF	Interventional (Sodium Chloride; Furosemide administration)	14	Urine exosome evaluation after treatment
17	NCT05370105	Recruiting	Stroke Rehabilitation	Observational	100	Characterization of blood EVs
18	NCT03264976	Not yet recruiting	DR	Observational	200	miRNA sequencing of serum exosomes
19	NCT04334603	Recruiting	HF	Interventional (Exercise training program)	120	Plasma exosomes concentration evaluation
20	NCT03227055	Unknown status	Childhood Chronic Kidney Disease	Observational	155	Urine exosome miRNAs evaluation
21	NCT04184076	Unknown status	Acute ischemic stroke	Interventional (Time-restricted feeding)	40	Characterization of plasma exosome markers (phospho-IRS1, phospho-Tau, Abeta1–42, phospho-IRS1)
22	NCT04641585	Not yet recruiting	Brugada Syndrome 1	Interventional (Brugada Syndrome diagnosti test)	144	Evaluation of exosome coding and noncoding RNAs
23	NCT05155358	Recruiting	Heat Stroke Early Waking MODS Proteinosis Liver Injury Kidney Injury	Observational	150	Plasma exosome miRNAs evaluation
24	NCT04235023	Unknown status	Sleep Apnea Inflammation Atherosclerosis	Interventional (apnea hypopnea index ≥ 15 per hour)	60	Plasma exosome evaluation
25	NCT03275363	Unknown status	Neurocognitive Disorder Mild Cognitive Impairment Alzheimer Dementia Vascular Dementia Age-related Cognitive Decline	Observational	500	Blood microvesicles and exosome analysis
26	NCT04266639	Completed	Stroke, Acute Ischemic Stroke Cerebrovascular Disorders Central Nervous System Diseases	Interventional (Remote ischemic conditioning)	45	Characterization of EVs surface markers and content after treatment
27	NCT05624203	Not yet recruiting	Myocardial Reperfusion Injury Treatment Outcome Prognosis ST Elevation Myocardial Infarction	Interventional (Exercise training)	100	Blood microvesicles analysis
28	NCT04250493	Recruiting	Multiple System Atrophy	Interventional (HOMA)	124	Evaluation of neural-derived plasma exosomes after HOMA

Abbreviations: ATAAD: Acute type A aortic dissection; AF: Atrial Fibrillation; CCU: Chronic skin ulcers; DASH: Dietary approaches to stop hypertension diet; DR: Diabetic retinopathy; EVs: Extracellular vesicles; HF: Heart Failure; HFpEF: Heart failure with preserved ejection fraction; HOMA: Homeostasis model assessment of insulin resistance fasting; MI: Myocardial infarction; MODS: Multiple organ dysfunction syndrome; MSC: Mesenchymal Stromal Cells; NCT: National Clinical Trial.

**Table 2 ijms-24-07509-t002:** Clinical studies of EV evaluation as diagnostic biomarkers for cancer treatment.

Number	NCT Number	Status	Conditions	Study Type	Population (Participants Number)	Outcome
1	NCT02702856	Completed	Prostate Cancer	Observational Multi center	2000	Validation of the association between exosome urine gene signature with high Gleason grade/score (GS ≥ 7)
2	NCT04529915	Active, not recruiting	Lung Cancer	Observational Multi center	470	Analysis of blood exosomes signature as an early marker of lung cancer
3	NCT04394572	Completed	Colorectal Cancer	Observational	80	Quantification of serum exosomes using specific exosomes markers as CD63 and CD81
4	NCT04939324	Recruiting	Lung Cancer NSCL Cancer	Interventional exosome will be collected from tumor drain vein during surgery	30	Analysis of molecular profiling of exosome with a sample in tumor-draining vein in order to identify prognostic molecular characteristics associated with cancer recurrence after surgery
5	NCT02393703	Recruiting	Pancreatic Cancer Benign Pancreatic Disease	Observational diagnostic	111	Differential gene expression in exosomes from cancer and healthy subjects to connect disease recurrence and outcomes with peculiar gene expression
6	NCT01294072	Recruiting	Colon Cancer	Interventional Curcumin delivery	35	Curcumin delivery by using plant exosomes to target the drug to colon tumors and normal colon tissue
7	NCT01159288	Completed	NSCL Cancer	Interventional cyclophosphamide (mCTX) followed by vaccinations with tumor antigen-loaded dendritic cell-derived exosomes (Dex).	41	Evaluation of tumor antigen-loaded in dendritic cell-derived exosomes
8	NCT01668849	Completed	Head and Neck Cancer Oral Mucositis	Interventional Chemoradiation Lortab, Fentanyl patch.	60	Investigate the ability of plant (grape) exosomes to prevent oral mucositis associated with chemoradiation
9	NCT04288141	Recruiting	HER2-positive Breast Cancer	Observational diagnostic	40	Evaluation of peripheral blood exosomes containing HER2 and HER3 dimers to predict tumor positivity
10	NCT05587114	Recruiting	Lung Cancer	Observational diagnostic	150	Exosome markers derived from peripheral blood and pulmonary venous blood from patients who underwent lung cancer surgery as early tumor markers
11	NCT03608631	Recruiting	Metastatic Pancreatic Adenocarcinoma Pancreatic Ductal Adenocarcinoma	Interventional Drug: Mesenchymal Stromal Cells-derived Exosomes with KRAS G12D siRNA	28	Evaluation of Exosomes with KRAS G12D siRNA in blood
12	NCT04653740	Recruiting	Advanced Breast Cancer	Interventional changes in the profile of tumor, associated with resistance to palbociclib at the individual level	25	Exosome characterization associated with palbociclib resistance
13	NCT03830619	Completed	Lung Cancer	Observational diagnostic	1000	Serum exosomal lncRNAs characterization
14	NCT04155359	Recruiting	Bladder Cancer	Observational diagnostic	3000	Evaluation of urine exosomes miR Sentinel™ BCa Test, a urine exosome-based diagnostic test, as an aid in diagnosing bladder cancer
15	NCT05427227	Recruiting	Advanced or Late Stage Gastrointestinal Cancer	Observational diagnostic	500	Plasma derived exosomes to explore the efficacy and mechanism of anti-HER2, immunotherapy and anti-CLDN18.2 of gastrointestinal cancer
16	NCT04167722	Recruiting	Prostate Cancer	Observational Robotic Radical Prostatectomy	100	Evaluation of exosomal small RNAs in lean vs. obese patients (array)
17	NCT04258735	Recruiting	Metastatic Breast Cancer	Interventional First line of treatment	300	Peripheral blood exosome nucleic acid content evaluation after first line of treatment
18	NCT03109873	Completed	Larynx, Lip, Oral Cavity, Pharynx Cancer	Interventional Drug: Metformin Hydrochloride	9	Characterization of exosome profile
19	NCT04629079	Recruiting	Lung Cancer	Observational Diagnostic	800	Characterization of exosomal pre-microRNA early detection in hypoxia tumor
20	NCT04556916	Recruiting	Prostate Cancer	Observational Diagnostic	320	Molecular characterization of circulating blood exosomes as diagnostic markers
21	NCT03895216	Completed	Bone Metastases	Observational Diagnostic	34	Evaluation of miRNA content of circulating tumor exosomes after surgery of bone metastasis
22	NCT02147418	Recruiting	Oropharyngeal Cancer	Observational Case control	30	Exosome testing as a screening modality for human papillomavirus-positive oropharyngeal squamous cell carcinoma
23	NCT04530890	Recruiting	Breast Cancer Digestive Cancer Gynecologic Cancer	Interventional Diagnostic	1000	Evaluation of the exosomes diagnostic value
24	NCT03032913	Completed	PDAC	Observational Diagnostic	52	Evaluation of onco-exosome quantification in the diagnosis of pancreatic cancer
25	NCT03488134	Active, not recruiting	Thyroid Cancer	Observational Diagnostic	74	Predicting prognosis and recurrence of thyroid cancer via new biomarkers, urinary exosomal Thyroglobulin and Galectin-3
26	NCT04453046	Terminated	Squamous Cell Carcinoma of the Head and Neck	Interventional Drug: Hemopurifier, Pembrolizumab	2	Kinetics of exosome depletion and reactivation in individual patient during the treatment
27	NCT02662621	Completed	Cancer	Interventional hormone therapy	71	Evaluation of HSP70-exosomes in blood and urine prognostic marker
28	NCT05424029	Recruiting	NSCL Cancer	Observational Diagnostic	200	Extracellular vesicles and particles (EVP) number as biomarkers of recurrence in non-small cell lung cancer
29	NCT04960956	Terminated	Prostate Cancer Urothelial Carcinoma	Observational Diagnostic	13	Evaluation of urine glycosylate exosomes
30	NCT04781062	Active, not recruiting	Breast Cancer	Interventional	367	MiRNA exosome sequencing As signature of early diagnosis
31	NCT04100811	Recruiting	Prostate Cancer	Observational Diagnostic	4000	SncRNAs urine exosome validation platform
32	NCT04720599	Completed	Urologic Cancer	Observational Diagnostic	120	ExoDx Prostate (IntelliScore) in men presenting for initial prostate biopsy
33	NCT03811600	Completed	Sleep Apnea Syndromes Obstructive Cancer	Observational Diagnostic	90	Evaluation of exosomal PD-1/PD-L1 expression
34	NCT03432806	Recruiting	Colon Cancer Liver Tumors	Observational Diagnostic	80	Characterization of exosomal proteins as biomarkers to monitor the liver pre-metastatic niche
35	NCT04357717	Terminated	Prostate Cancer	Observational Diagnostic early marker	150	Clinical performance of the ExoDx prostate cancer
36	NCT05559177	Recruiting	Chimeric exosomal tumor vaccines	Interventional Chimeric exosomal tumor vaccines	9	Dose evaluation of chimeric exosome vaccine
37	NCT04852653	Recruiting	Rectal Cancer, Liquid Biopsy	Observational	40	Evaluating extracellular vesicles obtained via liquid biopsy for neoadjuvant treatment response assessment in rectal cancer (RECC-EV)
38	NCT02977468	Recruiting	Triple Negative Breast Cancer	Interventional Drug: Pembrolizumab	15	Evaluation of serum exosomes as marker in response to therapy
39	NCT03824275	Active, not recruiting	Prostatic Neoplasms	Interventional Drug: ^18^F-DCFPyL PET/CT	129	Characterization of circulating exosomes
40	NCT03911999	Completed	Prostate Cancer	Observational	180	Identification of potential exosomal microRNAs in urine as marker of diagnosis
41	NCT05572099	Recruiting	Prostate Cancer	Observational	750	Evaluation of the ExoDx prostate test
42	NCT03711890	Recruiting	Pancreatic Carcinoma Pancreatic Intraductal Papillary Mucinous Neoplasm	Interventional procedure: Optical Coherence Tomography	75	Evaluation of blood cancer-derived exosomes
43	NCT03031418	Completed	Cancer of Prostate	Observational Validation diagnostic tool	532	Clinical Evaluation of the ‘ExoDx Prostate IntelliScore’
44	NCT03317080	Active, not recruiting	Lung Cancer	Observational Early diagnosis	400	Clinical evaluation of tumor exosome liquid biopsy
45	NCT05397548	Recruiting	Gastric Cancer	Observational Early diagnosis	700	Evaluation of predictive value of circulating exosomal lncRNA-GC1
46	NCT02862470	Completed	Thyroid Cancer	Observational Early diagnosis	22	Evaluation of urine exosomes as biological markers
47	NCT03791073	Recruiting	Oncology	Observational	200	Evaluation of interstitial tissue fluid as source of exosomes
48	NCT03235687	Active, not recruiting	Cancer of the Prostate	Interventional Surgery	1000	Evaluation of the ExoDx Prostate (IntelliScore)
49	NCT02063464	Completed	Ovarian Cancer Cancer of the Ovary Ovarian Neoplasms	Observational	85	Evaluation of blood exosomes as prognostic and outcome
50	NCT05375604	Recruiting	Advanced HCC Gastric Cancer Metastatic to Liver Colorectal Cancer Metastatic to Liver	Interventional Drug: CDK-004	30	Evaluation of CDK-004 as marker of cell-derived exosomes efficiency
51	NCT05101655	Completed	Osteosarcoma Pulmonary Metastases	Observational Diagnosis	60	Development of microfluidic chip technology to capture exosome subgroups
52	NCT03108677	Active, not recruiting	Lung Metastases Osteosarcoma	Observational Diagnosis	90	Identification of RNA profile from circulating exosomes
53	NCT02439008	Terminated	Carcinoma Hepatocellular Colorectal Neoplasms Melanoma Kidney Neoplasms	Interventional Radiotherapy	28	Quantification of secreted exosomes before and after radiotherapy
54	NCT05334849	Completed	Advanced Gastric Carcinoma Immunotherapy	Observational	80	Identification of identified circulating exosomal lncRNA-GC1 as marker of efficacy of immunotherapy
55	NCT03250078	Recruiting	Pancreatic Neoplasms	Observational	100	Evaluation of serum exosomes carrying gene mutation associate with high risk of hereditary gastric cancer
56	NCT03109847	Completed	Thyroid Cancer	Interventional Drug: Metformin Hydrochloride Radioactive iodine	13	Characterization of serum and salivary exosome profile
57	NCT02921854	Completed	NSCL Cancer	Interventional Immunotherapy	60	Detection of circulating markers of immunogenic cell death
58	NCT05192694	Recruiting	Prostate Cancer	Interventional Procedure: Tomography	40	Extracellular vesicles (exosomes) isolated and markers of reactive stroma
59	NCT02507583	Completed	Malignant Glioma Neoplasms	Interventional Drug: Exosomal IGF-1R/AS	33	Evaluation of blood IGF-1R/AS as dose escalation
60	NCT04948437	Recruiting	Sarcoma	Observational Diagnosis	103	Urinary exosomal thyroglobulin, galectin-3, calprotectin A9, transketolase, keratin 19, angiopoietin-1, tissue inhibitor of metalloproteinas, keratin 8, calprotectin A8, annexin II.
61	NCT03800121	Recruiting	Glioblastoma Multiforme	Observational Diagnosis	30	Evaluation of blood exosomes
62	NCT02071719	Terminated	Renal Cell Cancer	Observational	6	Evaluation of serum and urine exosomes
63	NCT01550523	Completed	Malignant Glioma of Brain	Interventional Drug: exosomal IGF-1R/AS ODN	13	Evaluation of exosomal IGF-1R/AS ODN-induced tumor cell
64	NCT02892734	Terminated	HER2/Neu Negative, Recurrent Inflammatory Breast Carcinoma Breast Cancer Breast Carcinoma	Interventional Drug: Ipililumab Nivolumab	3	Evaluation of blood exosomes as response to therapy
65	NCT04053855	Recruiting	Clear Cell Renal Cell Carcinoma	Observational Diagnosis	100	Evaluation of urinary exosomes
66	NCT05705583	Recruiting	Renal Cell Carcinoma	Observational	100	Evaluation of circulating exosomes blood and urine as marker of response to immunotherapy
67	NCT02928432	Completed	Prostate Cancer	Interventional Drug: Rednisone to dexamethasone change in mCRPC patients treated with abiraterone	26	Evaluation of blood exosomes as marker of therapy switch
68	NCT04499794	Recruiting	Untreated Advanced NSCL cancer Patients FISH Identified ALK Fusion Positive or Negative	Interventional Drug: ULK inhibitor	75	Study of exosomal EML4-ALK fusion
69	NCT03228277	Completed	NSCL Cancer	Interventional Drug: Olmutinib	25	Evaluation of DNA T790M mutation from bronchoalveolar lavage fluid EVs
70	NCT03334708	Recruiting	Pancreatic Cancer, Pancreatic Diseases, Pancreatitis, Pancreatic Cyst	Observational Diagnosis	700	Evaluation of blood exosome profile
71	NCT03985696	Recruiting	Lymphoma, B-cell, Aggressive Non-Hodgkin (B-NHL)	Interventional Drug: anti-CD20 and Nivolumab	90	Evaluation of exosomal CD20 and PD-L1 exosomes and resistance to immunotherapy
72	NCT04340245	Active, not recruiting	Prostate Cancer	Observational Diagnosis	60	Evaluation of blood and urine exosomes to discriminate progressors from non-progressors
73	NCT05575622	Recruiting	Hepatocellular Carcinoma	Observational Diagnosis	200	Clinical analysis of CTC and exosome combination and efficacy of immunotherapy in patients with hepatocellular carcinoma
74	NCT03854032	Active, not recruiting	Lip, Oral Cavity Squamous Cell Carcinoma, Pharynx, Larynx, Squamous Cell Carcinoma	Interventional Drug: Nivolumab	45	Evaluation of blood exosome abundance and composition and treating patients with stage II-IV squamous cell cancer
75	NCT03236675	Active, not recruiting	Carcinoma, Non-Small-Cell Lung	Observational Diagnosis	60	Evaluation of exosomal EML4-ALK fusion transcripts and T790M EGFR mutation
76	NCT01629498	Recruiting	Recurrent Lung Non-Small Cell Carcinoma	Interventional Image-Guided	100	Evaluation of blood exosomes and treatment of regionally metastatic head and neck squamous cell cancer
77	NCT05110781	Recruiting	Cutaneous Squamous Cell Carcinoma of the Head and Neck	Interventional Drug: Atezolizumab	18	Evaluation of plasma exosomal levels
78	NCT05441189	Completed	Stage I–II PDAC	Observational Diagnosis	70	Evaluation of specific exosome signature and gene signature for predicting
79	NCT05166616	Recruiting	Advanced NSCL Carcinoma	Interventional Drug: Minnelide and Osimertinib	30	Evaluation of exosomal pattern and treatment of advanced EGFR mutated non-small cell lung cancer
80	NCT03096340	Terminated	Cancer	Observational Drug: IT-141	10	Evaluation of exosomes presence and monotherapy in patients with advanced cancer
81	NCT00578240	Active, not recruiting	Prostate Cancer	Observational Diagnosis	5290	Evaluation of exosome presence into peripheral blood, bone marrow, urine and other bodily fluids
82	NCT02051101	Completed	Port-Wine Stain	Interventional	19	Characterization of exosome profile and pathogenic mechanisms of port wine stain
83	NCT02535247	Terminated	Lymphoma T-Cell, Peripheral	Interventional Drug: MK-3475	17	Circulating exosome analysis and study of MK-3475 R/R NK and T-cell non-Hodgkin Lymphoma
84	NCT03410030	Completed	Pancreatic Adenocarcinoma Resectable and Metastases	Interventional Drug: AA NABPLAGEM	27	Evaluation of potential exosome pattern as biomarker and AA NABPLAGEM
85	NCT03083678	Active, not recruiting	Chordoma	Interventional Drug: Afatinib	43	Evaluation of circulating exosomes and Afatinib in locally advanced and metastatic chordoma
86	NCT03217266	Active, not recruiting	Soft Tissue Sarcoma	Interventional Drug: Navtemadlin	46	Evaluation of tumor genetic mutations in deoxyribonucleic acid ribonucleic acid isolated from exosomes and Navtemadlin and radiation therapy in treating patients with soft tissue sarcoma
87	NCT03537599	Completed	Recurrent Acute Myeloid Leukemia with Myelodysplasia	Interventional Drug: Daratumumab	4	Evaluation of exosome pattern and Daratumumab and donor lymphocyte infusion in treating participants with relapsed acute myeloid leukemia after stem cell transplant
88	NCT04483219	Recruiting	Metastatic Colorectal Adenocarcinoma	Interventional Drug: Tyrosine Kinase Inhibitor (TKI) + Anti-PD-1 Antibody	53	Evaluation of circulating exosome pattern in metastatic colorectal adenocarcinoma

Abbreviations: NSCL: Non-Small Cell Lung; lncRNAs: long noncoding RNAs; sncRNAs: small noncoding RNAs; HCC: Hepatocellular Carcinoma; PDAC: pancreatic ductal adenocarcinoma; miRNA: micro RNA; NCT: National Clinical Trial.

## Data Availability

Not applicable.

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
