# Peer review of "Basic Pathogenic Mechanisms and Epigenetic Players Promoted by Extracellular Vesicles in Vascular Damage"

_ijms, 2023, doi:10.3390/ijms24087509_

Round 1

Reviewer 1 Report

Briefly, Authors provide a broad overview on the role of extracellular vesicles in the epigenetic-mediated pathogenesis of vascular damage in cardiovascular diseases, and in the neoangiogenesis occurring in tumor progression. In general, the review is well structured. However, I would suggest some changes in order to make it more readable and understandable.

-          2. “EVs biology”.

This reviewer suggests providing a more detailed explanation of the extracellular vesicles release mechanism. Furthermore, including some references on the methods used for vesicles isolation would make the manuscript more comprehensive.

-          3. “The epigenetic Role of EVs in cardiovascular damage”.

·         Line 96: this reviewer suggests deleting the full stop “.” if it was a mistake.

·         Line 152: this reviewer suggests deleting the quotation marks if it was a mistake.

·         Lines 152-157: this reviewer suggests describing more in detail the study cited by clarifying the effect produced by epigenetic changes. 

-           4. “Basic mechanisms and epigenetic role of EVs in tumor progression”

·         Line 262: this reviewer suggests writing in full the abbreviation UFC1 the first time it occurs in the text. This should be applied to all abbreviations in the manuscript, where possible.

-        Authors have mostly discussed the role of non-coding RNAs as epigenetic modifiers transported by extracellular vesicles. As introduced in the lines 59-62, extracellular vesicles cargo may include other epigenetic enzymes such as DNA methylase and histone deacetylases, barely addressed in the manuscript. Therefore, this reviewer suggests describing the contribute of these enzymes both in cardiovascular disease and in the tumor progression, if supported by the literature.   

-          Conclusion and perspectives

Conclusions should be improved. It is essential to emphasize the role of extracellular vesicles in epigenetic regulation within the context of cardiovascular pathology and cancer. Moreover, highlighting the potential clinical applications of these vesicles and the related innovative therapeutic strategies would strengthen the aim of this manuscript.

Author Response

Manuscript ID: ijms-2326347

Type of manuscript: Review

Title: Basic pathogenic mechanisms and epigenetic players promoted by extracellular vesicles in vascular damage

Reviewer1

Comments and Suggestions for Authors

Briefly, Authors provide a broad overview on the role of extracellular vesicles in the epigenetic-mediated pathogenesis of vascular damage in cardiovascular diseases, and in the neoangiogenesis occurring in tumor progression. In general, the review is well structured. However, I would suggest some changes in order to make it more readable and understandable.

  1. “EVs biology”.

This reviewer suggests providing a more detailed explanation of the extracellular vesicles release mechanism. Furthermore, including some references on the methods used for vesicles isolation would make the manuscript more comprehensive.

As required, we discussed extracellular biology and methods of preparation in section “EVs biology and function” and added references 26-29 and 30-32.

  1. The epigenetic Role of EVs in cardiovascular damage”. 

Line 96: this reviewer suggests deleting the full stop “.” if it was a mistake.

As required, it was deleted.

Line 152: this reviewer suggests deleting the quotation marks if it was a mistake.

As required, it was deleted.

Lines 152-157: this reviewer suggests describing more in detail the study cited by clarifying the effect produced by epigenetic changes.

As suggested, we modified the text as follows: Notably, ACS-derived EVs absorbed on PBMCs from healthy subjects modulated the expression of several genes, presumably via epigenetic mechanism in agreement with previous reports [62-66]. In particular EVs induced methylome changes promoting a significant activation of SYP, CHL1, and SHB genes [61]. These findings underlined the epigenetic contribution in the role of EVs in ACS.

  1. “Basic mechanisms and epigenetic role of EVs in tumor progression”

Line 262: this reviewer suggests writing in full the abbreviation UFC1 the first time it occurs in the text. This should be applied to all abbreviations in the manuscript, where possible.

We now provide the full names of the abbreviations throughout the text.

Authors have mostly discussed the role of non-coding RNAs as epigenetic modifiers transported by extracellular vesicles. As introduced in the lines 59-62, extracellular vesicles cargo may include other epigenetic enzymes such as DNA methylase and histone deacetylases, barely addressed in the manuscript. Therefore, this reviewer suggests describing the contribute of these enzymes both in cardiovascular disease and in the tumor progression, if supported by the literature.

We thank for the indication. The contribution of exosomal DNA methylase and histone deacetylases is not supported roboustly by the literature. We now state in the introduction that these enzymes play a key role in epigenetic modification, however, mainly miRNAs and long noncoding RNA are transported by EVs on endothelial cells (ECs) modifying DNA methylation status and gene expression.

  1. Conclusion and perspectives

Conclusions should be improved. It is essential to emphasize the role of extracellular vesicles in epigenetic regulation within the context of cardiovascular pathology and cancer. Moreover, highlighting the potential clinical applications of these vesicles and the related innovative therapeutic strategies would strengthen the aim of this manuscript.

As requested, we have modified the conclusions and, among other, now discuss cancer and CVDs common pathways in cancer and CVD that are activated by EVs.

Reviewer 2 Report

It is very unique and interesting to see discussion on roles of EVs and EV-miRNAs in vascular disease and cancer at the same time, showing very informative summary of clinical studies about therapeutic approaches found in “National Clinical Trial”.

Comments:

1. The author starts to discuss on roles of EVs for atherosclerosis, heart disease and cancers. This sounds as a kind of sudden introduction. EVs are playing roles for vascular damage in many other pathological status such as infectious disease, autoimmune diseases etc. It is better to give comprehensive introdcution of EVs role at first, then to focus on atherosclerosis and cancers.

2. It is not clear why this review contain two different topics, cardiovascular diseases and cancers. Some rearrangement of sentences may help reader’s understanding.

3. No explanation of “NCT” at all. Probably “National Clinical Trial” it is.

Table 1 & 2 showing 28 NCT in cardiovascular and 88 in oncology field are only mentioned in “Conclusion and perspectives”. There is no consistency with comprehensive description in sections 3 and 4. It is more impressive to connect basic finding to NCTs.

4. The authors should give some comments on non-EV miRNAs for biological function or as possible biomarkers, since more than 50% of circulating miRNAs are not inside EVs.

5. Reference 69 lacks description of the paper. (Line 555)

6. In discussion, the authors gave comprehensive summary of broad data obtained from basic and clinical observations. It is very nice to have perspective view of biological role of EV-miRNAs and lncRNAs. However, mechanisms of cardiovascular damages caused by EV-miRNAs were not efficiently correlated to tumor progression or metastasis. I feel some gap between two fields. It is better to emphasize more on common EV-related mechanisms of two fields.

Author Response

Reviewer2

Comments and Suggestions for Authors

It is very unique and interesting to see discussion on roles of EVs and EV-miRNAs in vascular disease and cancer at the same time, showing very informative summary of clinical studies about therapeutic approaches found in “National Clinical Trial”.

Comments:

  1. The author starts to discuss on roles of EVs for atherosclerosis, heart disease and cancers. This sounds as a kind of sudden introduction. EVs are playing roles for vascular damage in many other pathological status, such as infectious disease, autoimmune diseases etc. It is better to give comprehensive introduction of EVs role at first, then to focus on atherosclerosis and cancers.

We thank for the indication. Accordingly, we modified the EVs biology section introducing the role of EVS in different physiological and pathological conditions and added references 26-29 and 30-32

  1. It is not clear why this review contains two different topics, cardiovascular diseases and cancers. Some rearrangement of sentences may help reader’s understanding.

As in our previous study (Sarno et al, Ref.18), both CVD and cancer shared common paths in vascular damage. We better explain this point as follows: the vasculature is sensitive to microenvironment stimuli and modifies its phenotype and behavior, including cell proliferation, migration and inflammation. These changes affect both tumors and CVD. We discuss this in the introduction and expanded the discussion of phenotype modification induced by exosomal activating epigenetic pathways through the text.

  1. No explanation of “NCT” at all. Probably “National Clinical Trial” it is.

Table 1 & 2 showing 28 NCT in cardiovascular and 88 in oncology field are only mentioned in “Conclusion and perspectives”. There is no consistency with comprehensive description in sections 3 and 4. It is more impressive to connect basic finding to NCTs.

We rephrased the text as indicated and now discuss the clinical trials in corresponding section.

  1. The authors should give some comments on non-EV miRNAs for biological function or as possible biomarkers, since more than 50% of circulating miRNAs are not inside EVs.

We thank the Reviewer for the interesting clarification to increase the quality of the manuscript. As requested, we now discussed the differences between non-EV miRNAs and EV miRNAs and putative utility in the conclusion paragraph.

  1. Reference 69 lacks description of the paper (Line 555)

Done.

  1. In discussion, the authors gave comprehensive summary of broad data obtained from basic and clinical observations. It is very nice to have perspective view of biological role of EV-miRNAs and lncRNAs. However, mechanisms of cardiovascular damages caused by EV-miRNAs were not efficiently correlated to tumor progression or metastasis. I feel some gap between two fields. It is better to emphasize more on common EV-related mechanisms of two fields.

As indicated by the Reviewer, in order to observe if the mechanisms of cardiovascular damages caused by EV-miRNAs were correlated to tumor progression or metastasis, we evaluated the single factors in both fields. Interestingly, we found that only 3 ncRNAs, specifically, miR-92a-3p, miR-145, and miR-210 were involved in vascular damage both in the context of CVDs and in tumor neoangiogenesis, although by different mechanisms. We discuss this point in the conclusion.